# Ferroptosis Regulated by Hypoxia in Cells

**DOI:** 10.3390/cells12071050

**Published:** 2023-03-30

**Authors:** Xiangnan Zheng, Yuqiong Liang, Cen Zhang

**Affiliations:** College of Biological Science and Engineering, Fuzhou University, Fuzhou 350108, China

**Keywords:** ferroptosis, hypoxia, iron metabolism, lipid peroxidation, hypoxia-inducible factors

## Abstract

Ferroptosis is an oxidative damage-related, iron-dependent regulated cell death with intracellular lipid peroxide accumulation, which is associated with many physiological and pathological processes. It exhibits unique features that are morphologically, biochemically, and immunologically distinct from other regulated cell death forms. Ferroptosis is regulated by iron metabolism, lipid metabolism, anti-oxidant defense systems, as well as various signal pathways. Hypoxia, which is found in a group of physiological and pathological conditions, can affect multiple cellular functions by activation of the hypoxia-inducible factor (HIF) signaling and other mechanisms. Emerging evidence demonstrated that hypoxia regulates ferroptosis in certain cell types and conditions. In this review, we summarize the basic mechanisms and regulations of ferroptosis and hypoxia, as well as the regulation of ferroptosis by hypoxia in physiological and pathological conditions, which may contribute to the numerous diseases therapies.

## 1. Introduction

Ferroptosis is a novel characterized oxidative damage-related regulated cell death, which is mainly driven by iron accumulation, lipid peroxidation, and subsequent plasma membrane rupture [1,2]. As a newly discovered form of regulated cell death, ferroptosis is morphologically, chemically, and genetically different from apoptosis, autophagy, necroptosis, and pyroptosis, and is regulated by its unique metabolism and regulatory mechanism [1,3]. With the increasing studies in recent years, more and more evidence demonstrated that ferroptosis may play a significant role in multiple physiological and pathological processes, such as embryonic development, cancer, neurodegeneration disease, and organ disorders caused by drug-induced toxicity or ischemia/reperfusion injury (IRI) [4,5,6,7,8,9]. In cancer, ferroptosis is reported to play an essential role in suppressing spontaneous tumorigenesis in mouse models [10,11]; many cancer cells resistant to conventional therapies were found to be susceptible to ferroptosis [12,13], suggesting that the induction of ferroptosis in cancer cells is a promising anti-tumor strategy. In neurodegeneration disease and the injuries of many organs such as the heart, brain, kidney, and liver, ferroptosis was identified as an important form of cell death that results in organ damage [7,8,9], suggesting that inhibiting ferroptosis could be an important way to limit organ damage in disorders. It seems that ferroptosis is regulated in context-dependent mechanisms, which may be important for developing therapeutic strategies for these diseases by targeting ferroptosis.

Low levels of oxygen in tissues, called hypoxia, can affect multiple cellular functions across a variety of cell types, and is associated with both physiological and pathological conditions, such as high altitude, solid tumors, and ischemia of organs [14,15,16]. Hypoxia is caused by many potential reasons in humans, such as insufficient blood flow to a specific area, decrease in hemoglobin levels, or treatment with chemical compounds [17,18]. Under hypoxic conditions, cells activate the hypoxia signaling pathway, which is mainly mediated by the hypoxia-inducible factors (HIFs), to induce a series of cellular responses to help cells to adapt to or escape from the hypoxic environment [19,20,21]. Interestingly, hypoxia is often associated with the increase in reactive oxygen species (ROS) and oxidative stress [22,23]; therefore, ferroptosis is surprisingly regulated by hypoxia in certain cells and conditions. In the past several years, many studies have demonstrated that there is a complicated network for the regulation of ferroptosis by hypoxia in different physiological and pathological conditions, which is summarized in this review.

## 2. Ferroptosis and Its Mechanism

Ferroptosis was first introduced in a study on the mechanism of some oncogenic RAS-selective lethal small molecule compounds, such as erastin and RSL3 in 2012 [24]. These compounds trigger a kind of cell death, termed ferroptosis, which cannot be blocked by the specific inhibitors for apoptosis, necrosis, necroptosis, and autophagy (e.g., Z-VAD-FMK, BOC-D-FMK, wortmannin, and necrostatin-1), but can be blocked by antioxidants (e.g., vitamin E), scavengers for lipid peroxidation (e.g., ferrostatin-1 or liproxstatin-1), and iron chelators (e.g., deferoxamine mesylate). Distinct from apoptosis, autophagy, and necroptosis, ferroptosis has its characteristic morphological, biochemical, and immunological features. Morphologically, ferroptotic cells generally exhibit necrosis-like changes, including plasma membrane integrity loss, swelling of cytoplasmic and cytoplasmic organelles, rupture of the mitochondrial outer membrane, reduced or absent mitochondrial crista, and moderate chromatin condensation [2,25]. However, there are no apoptotic bodies or autophagosomes in ferroptotic cells, which are typically found in apoptosis and autophagy, respectively. Biochemically, ferroptosis is characterized by iron accumulation and lipid peroxidation, as well as ROS and oxidative stress [1,2,25]. Immunologically, ferroptosis is considered as a form of inflammatory cell death associated with the release of damage-associated molecular pattern (DAMP) or lipid oxidation products [2].

Although the molecular mechanism of ferroptosis is not fully understood, it is universally accepted that ferroptosis is caused by abnormally increased iron accumulation and lipid peroxidation, as well as the dysregulation of antioxidant defense systems (Figure 1).

### 2.1. Iron Metabolism

In cells, iron mainly exists in two states, ferrous (Fe^2+^) and ferric (Fe^3+^). Ferroptosis is promoted by intracellular free Fe^2+^ in various intracellular iron pools, including lysosomes and endoplasmic reticulum (ER), through the Fenton reaction that directly generates ROS, or the activation of iron-containing enzymes such as lipoxygenase that catalyze the lipid peroxidation [2,6]. The intracellular free Fe^2+^ level is controlled by an integrated system affecting the absorption, storage, efflux, and utilization of iron. The dysregulation of any step in this system may affect the ferroptosis [2,6].

The Fe^2+^ generated through intestinal absorption or degradation by red blood cells is oxidized to Fe^3+^ in the blood [26]. In the blood, Fe^3+^ binds to transferrin (TF) or lactotransferrin (LTF) and is then recognized by the transferrin receptor (TFRC) in the cell membrane [26,27]. After absorption by endocytosing of the TF-TFRC complex, Fe^3+^ is reduced to Fe^2+^ in the endosome by ferrireductase activity of six-transmembrane epithelial antigens of the prostate 3 (STEAP3) in the endosome [28]. Finally, the reduced Fe^2+^ is released into the free iron pool in the cytoplasm by divalent metal transporter 1 (DMT1, also known as solute carrier family 11 member 2, SLC11A2) [26,27]. Elevation of TF, LTF, TFRC, STEAP3, or DMT1 can enhance iron uptake to promote ferroptosis [2,27]. Additionally, two metal-ion transporters of zinc-regulated transporters, iron-regulated transporter-like proteins (ZIP) family, ZIP8 and ZIP14, can mediate the absorption of iron [29]. Heat shock protein family B (small) member 1 (HSPB1) phosphorylated by protein kinase C (PKC) was also reported to reduce the iron uptake through regulating the cytoskeleton organization and thereby inhibiting ferroptosis [30]. Alternatively, iron can also be absorbed as heme and be released to the cytoplasm through degradation by heme oxygenase 1 (HO-1) or HO-2. Thus, the ferroptosis will also be regulated by HO-1 and HO-2 [31,32].

Excess intracellular Fe^2+^ can be stored by ferritin, a protein complex composed of ferritin light chain (FTL) and ferritin heavy chain 1 (FTH1), to prevent Fe^2+^ from being oxidized [27]. Nuclear receptor coactivator 4 (NCOA4) binds to ferritin and then transports iron-bound ferritin to the autophagosome for lysosomal degradation (also known as ferritinophagy), leading to the release of Fe^2+^ [33]. Thus, increasing the expression of *FTL* and *FTH1*, or the inhibition of NCOA4, will reduce the intracellular free Fe^2+^ and inhibit ferroptosis [27,33]. In addition to ferritin, the overexpression of ferritin mitochondrial (FTMT), another iron-storage protein in mitochondria, leads to the decrease in intracellular free Fe^2+^ and inhibits ferroptosis [34].

The intracellular Fe^2+^ can be exported by iron-efflux protein ferroportin (FPN1, also known as solute carrier family 40 member 1, SLC40A1) and then be reoxidized to Fe^3+^ [35]. The internalization and degradation of FPN1 are induced by hepcidin (encoded by the *Hamp1* gene), a peptide hormone regulated by the BMP/Smad pathway [36]. The decrease in FPN1 or elevation of hepcidin levels releases the free Fe^2+^ to promote ferroptosis [27].

Additionally, iron was used for biogenesis of some mitochondrial proteins with iron-sulfur cluster, such as NFS1, CISD1, and CISD2, leading to the decrease in the available iron levels to inhibit ferroptosis [37,38,39].

### 2.2. Lipid Metabolism

Lipid peroxidation is the most important hallmark and plays a key role in ferroptosis [2,6]. Lipid peroxidation occurs through both non-enzymatically spontaneous autoxidation and enzyme-mediated processes [40,41]. Enzymatical lipid peroxidation is regulated by lipoxygenases (LOXs), and the suppression of LOXs levels or activities leads to ferroptosis inhibition in certain cell lines [42]. The arachidonate lipoxygenase (ALOX) family is a class of non-heme iron-containing enzymes that can catalyze polyunsaturated fatty acids (PUFAs) oxygenation to generate various hydroperoxy PUFA derivatives, such as the initial lipid hydroperoxides and subsequent reactive toxic aldehydes, such as malondialdehyde (MDA) and 4-hydroxynonenal (4-HNE). The mammalian ALOX family, including six members (ALOXE3, ALOX5, ALOX12, ALOX12B, ALOX15, and ALOX15B), play a tissue- or cell-dependent role in mediating ferroptosis [40,41]. Cyclooxygenase-2 (COX-2, encoded by the *PTGS2* gene), which oxidizes lysophospholipids but not phospholipids, is mainly considered as a biomarker of ferroptosis [2,43]. However, it is also reported in some cases that COX-2 can mediate lipid peroxidation and induce ferroptosis [44,45]. Alternatively, membrane electron transfer proteins, such as cytochrome P450 oxidoreductase (POR) and nicotinamide adenine dinucleotide phosphate (NADPH) oxidase (NOX), contribute to non-enzymatic lipid peroxidation through ROS production in ferroptosis [46,47,48].

PUFA production for subsequent lipid peroxidation is supported by upstream lipid synthesis and metabolism pathways. PUFA-phospholipids, especially arachidonic acid (AA) and adrenic acid (AdA), are the main substrates of lipid peroxidation in ferroptosis [40,41]. Acyl-CoA synthetase long-chain family member 4 (ACSL4) catalyzes the ligation reaction of CoA with AdA/AA to form CoA-AdA/AA intermediate, which subsequently undergoes esterification with membrane phosphatidylethanolamine to form PE-AdA/AA by lysophosphatidylcholine acyltransferase 3 (LPCAT3). The loss of either ACSL4 or LPCAT3 results in the resistance to ferroptosis [49]. Meanwhile, monounsaturated fatty acids (MUFAs) that lack the bis-allylic positions readily for peroxidation, inhibit ferroptosis by competing with PUFAs. The inhibition of ferroptosis by MUFAs relies on ACSL3, which catalyzes the ligation reaction of CoA with MUFAs, or stearoyl-CoA desaturase 1 (SCD1), the rate-limiting enzyme in MUFA production [50,51].

The processes of lipid synthesis, absorption, storage, and release are also involved in ferroptosis. Long-chain PUFAs, including AA and AdA, are synthesized from dietary essential fatty acids in cells by a series of enzymatic reactions involving the elongation of very long-chain fatty acid protein (ELOVL) and fatty acid desaturase (FADS). The silencing or inhibition of ELOVL5, FADS1, and FADS2 was shown to inhibit ferroptosis [52,53]. β-oxidation, the process in which fatty acids are broken down to produce acetyl-CoA, is generally believed to negatively regulate ferroptosis by decreasing the availability of unesterified PUFAs [41]. Fatty acids are absorbed into cells through various fatty acid transport proteins, including fatty acid translocase (FAT/CD36), fatty acid transport proteins (FATPs), and fatty acid-binding proteins (FABPs) [54]. CD36 and FATP2 mediate the absorption of AA and AdA to promote ferroptosis in certain cells [55,56]. In addition to lipid uptake, FABPs also facilitate the transport of lipids to specific compartments in the cell, such as to the lipid droplet for lipid storage [57]. FABP3, FABP4, and FABP7 were reported to inhibit ferroptosis through enhancing fatty acid uptake and lipid storage with lipid droplet formation [58,59]. Moreover, increased lipid storage by tumor protein D52 (TPD52) represses lipid peroxidation and ferroptosis, whereas degradation of lipid droplets by autophagy (known as lipophagy) enhances free fatty acids production, and increases lipid peroxidation and ferroptosis [60]. Hypoxia-inducible, lipid droplet-associated protein (HILPDA), can increase PUFAs incorporation into triacylglycerols and phospholipids through binding to and inhibiting adipose triglyceride lipase, responsible for breaking down triacylglycerols [61]. During the continuous membrane remodeling, PUFAs are released from membrane phospholipids through hydrolysis catalyzed by phospholipase A2 (PLA_2_). PLA2G6 (also known as iPLA_2_β), a Ca^2+^-independent PLA_2_, was reported to inhibit ferroptosis by cleaving oxidized hydroperoxy-arachidonoyl (C20:4)- or adrenoyl (C22:4)- phosphatidylethanolamine (Hp-PE) from membrane phospholipids [62].

### 2.3. Antioxidant Defense System

Ferroptosis is caused by lipid peroxidation, which is inhibited by the antioxidant defense system in cells. Glutathione peroxidase 4 (GPX4) is a selenocysteine-containing enzyme that plays a central role in ferroptosis inhibition, which inhibits lipid peroxidation by directly reducing toxic phospholipid hydroperoxides (PL-OOH) to non-toxic phospholipid alcohols (PL-OH) by oxidizing glutathione (GSH) [63,64]. GPX4 dysfunction always results in the uncontrolled accumulation of lipid peroxides and ferroptosis. A group of small-molecule compounds, including RSL3, ML162, ML210, FIN56, and FINO_2_, can directly or indirectly inhibit GPX4 activity as well as promote the GPX4 protein degradation, to induce ferroptosis [63,64,65,66]. GPX4 is transcriptionally upregulated by the transcription factor AP-2 gamma (TFAP2C) and specificity protein 1 (SP1), which leads to ferroptosis inhibition [67]. As a selenocysteine-containing protein, GPX4 levels are regulated by the selenocysteine tRNA. Selenocysteine tRNA is positively regulated by isopentenyl pyrophosphate (IPP), which is a product of the mevalonate pathway for lipid synthesis [68]. Furthermore, GPX4 is also regulated by SCD1/FADS2, leading to the ferroptosis inhibition [69]. It seems that GPX4 is also regulated by lipid metabolism.

The reduction reaction catalyzed by GPX4 requires GSH, which is synthesized from glutamate, cysteine, and glycine [1,2,6]. Thus, ferroptosis is also regulated by the amino acid metabolism involved in GSH synthesis. System Xc^−^, a glutamic acid/cystine antiporter in the plasma membrane that is constituted by the transport subunit solute carrier family 7 member 11 (SLC7A11) and regulatory subunit solute carrier family 3 member 2 (SLC3A2), is responsible for the cystine uptake and plays a crucial role in ferroptosis repression [70]. Several small-molecule compounds (e.g., erastin, sulfasalazine, and sorafenib), have been shown to be the inhibitors of system Xc^−^ mediated cystine uptake, which decreases the GSH synthesis to trigger ferroptosis [65,70]. Beclin-1, an autophagy-associated protein, interacts with SLC7A11 when phosphorylated at Ser90/93/96 by AMPK, leading to the inhibition of system Xc^−^ to induce ferroptosis [71,72,73,74]. The cystine uptaken by system Xc^−^ is further reduced to cysteine by GSH and/or thioredoxin reductase 1 (TXNRD1) and is used for GSH synthesis by glutamate cysteine ligase (GCL) and glutathione synthetase (GSS). Inhibition of GCL or GSS, or activation of ChaC glutathione specific gamma-glutamylcyclotransferase 1 (CHAC1), an enzyme that catalyzes the GSH degradation, leading to ferroptosis induction [24,65,75,76].

In addition to the GPX4-GSH system, lipid peroxidation and ferroptosis were also found to be inhibited by ferroptosis suppressor protein 1 (FSP1, previously known as apoptosis-inducing factor mitochondria-associated 2, AIFM2). FSP1 can reduce non-mitochondrial coenzyme Q10 (CoQ10) using nicotinamide adenine dinucleotide phosphate (NADPH), thereby trapping lipid peroxides in a GPX4-independent manner and inhibiting ferroptosis [77,78]. It is also reported that FSP1 can repress ferroptosis through the activation of the ESCRT-III-dependent membrane repair system independent of its oxidoreductase function [79]. Tetrahydrobiopterin (BH_4_) produced by GTP cyclohydrolase-1 (GCH1) acts as a radical-trapping antioxidant to selectively prevent two PUFA acyl tails from consuming phospholipids and protect lipid membranes from ferroptosis [80]. Dihydrofolate reductase (DHFR) inhibits ferroptosis through regenerating oxidized BH4 [81]. Dihydroorotate dehydrogenase (DHODH) can convert CoQ10 to ubiquinol (a radical-trapping antioxidant with anti-ferroptosis activity) in mitochondria, to restore peroxide-damaged mitochondrial lipids, leading to ferroptosis inhibition [82]. A recent study reported that tryptophan metabolites serotonin (5-HT) and 3-hydroxyanthranilic acid (3-HAA) work as radical-trapping antioxidants to eliminate lipid peroxidation, thereby inhibiting ferroptosis [83].

Moreover, many proteins, especially transcription factors, regulate ferroptosis by directly or indirectly modulating a group of the above-mentioned genes or proteins. For example, nuclear factor erythroid 2-related factor 2 (NRF2, also known as nuclear factor erythroid-derived 2-like 2, NFE2L2), a master regulator of oxidative stress signaling, inhibits ferroptosis by transcriptionally regulating the expression of *FPN1*, *HO-1*, *FTL*, and *FTH1* involved in iron metabolism as well as *SLC7A11*, *TXNRD1*, *GSS*, *GCLC*, *GCLM*, *CHAC1*, *GPX4*, *FSP1*, and *NQO1* (quinone oxidoreductase 1) involved in antioxidant systems [84]. Tumor suppressor p53, a transcription factor that regulates the expression of stress response genes and mediates a variety of anti-proliferative processes, plays a dual role in ferroptosis [85]. p53 induces ferroptosis through transcriptionally regulating *SLC7A11* for the anti-oxidant defense, *PTGS2*, *ALOX12* for lipid metabolism, *FDXR* for iron metabolism, and *GLS2* (glutaminase 2, an enzyme for glutaminolysis that is involved in GSH synthesis and lipid synthesis) for both anti-oxidant system and lipid metabolism [85]. p53 also transcriptionally induces SAT1 (Spermidine/spermine N1-acetyltransferase 1) to elevate ALOX15 levels, leading to lipid peroxidation and ferroptosis [85,86]. On the contrary, p53 was also found to inhibit ferroptosis by transcriptionally inducing *p21* to enhance GSH levels and *Parkin* to induce mitophagy, as well as by directly binding to the dipeptidyl peptidase DPP4 to inhibit NOX-mediated lipid peroxidation [85]. Interestingly, HIFs, acting as a transcription factor for hypoxia, also play a complex role in ferroptosis, which will be reviewed later.

## 3. Hypoxia and Hypoxia-Induced Factors

Hypoxia refers to a limited oxygen level in tissues, which is caused by a variety of mechanisms in both physiological and pathological conditions. For instance, hypoxia in tumors can be caused by the abnormal disorganization of tumor vasculature (perfusion hypoxia), long oxygen diffusion distances (>70 μm) from blood vessels to tumor cells (diffusion hypoxia), or a reduced oxygen transport capacity resulting from chemotherapy-induced anemia (anemic hypoxia) [17,18]. These hypoxia conditions lead to the proliferation, angiogenesis, metastasis, metabolism reprogramming, stemness, immune evasion, and therapy resistance of tumor cells, and ultimately contribute to tumor progression [14,20,87,88]. During surgical operation or drug treatment in some organs, ischemia/reperfusion (I/R) results in the hypoxia/reoxygenation (H/R) of cells because of the changes in oxygen supply from the bloodstream, which finally leads to cell death and injury [16,89]. Moreover, hypoxic placenta resulting from the dysregulation of angiogenic factors, induces oxidative stress, lipid peroxidation, endothelial dysfunction, and peripheral vasoconstriction, leading to abnormal trophoblastic invasion and preeclampsia, which is a serious and distinct type of pregnancy-induced hypertension [13]. Generally, hypoxia is an important environmental factor in many physiological and pathological conditions.

In response to the hypoxic condition, cells activate the transcription machinery to modulate the expression of a group of genes to adapt to environmental change. HIFs, a family of transcription factors, play a central role in hypoxia response [19,20,21]. HIFs, including HIF-1, HIF-2, and HIF-3, are heterodimers constituted by one of three oxygen-dependent α-subunit (HIF-1α/HIF-2α/HIF-3α) and a constitutively expressed β-subunit (HIF-1β, also known as aryl hydrocarbon receptor nuclear translocator, ARNT). HIF-1α is the most well-studied member of the HIF family, which is ubiquitously expressed in most mammalian cells [20,21]. HIF-1α belongs to the basic-helix-loop-helix (bHLH)-PER (Period)-ARNT-SIM (single-minded) (PAS) superfamily, which contains bHLH and PAS domains for DNA binding and dimerization, respectively [90]. Under the normoxic condition, HIF-1α is hydroxylated at two proline residues, Pro402 and Pro564, in its oxygen-dependent degradation domain (ODD) by the prolyl hydroxylases (PHDs), resulting in its binding to the E3 ubiquitin ligase von Hippel-Lindau protein (pVHL) via its Leucine-574 residue (Leu574) for ubiquitination and degradation [20,21]. PHDs are dioxygenases using Fe^2+^ and ascorbate as cofactors, which depend on O_2_ as a direct substrate. Therefore, the hypoxic condition will inactivate PHDs, blocking the interaction between HIF-1α and pVHL, and eventually leading to HIF-1α stabilization and nuclear translocation [19,20,21]. In the nucleus, HIF-1α forms a heterodimer (HIF-1) with HIF-1β, which binds to E-box-like hypoxia response elements (HREs, containing the core sequence (A/G)CGTG), leading to the transcription of its target genes in association with the co-activator CREB-binding protein (CBP)/p300 [20,21]. HIF-1α is also hydroxylated at the asparagine-803 residue (at Asn851) under the normoxic condition by the factor inhibiting HIF-1 (FIH-1), another O_2_-dependent dioxygenase requiring Fe^2+^ and ascorbate as cofactors, which blocks the transcriptional activities of HIF-1α through disruption of its interaction with CBP/p300 [19,20,21]. HIF-2α and HIF-3α have high sequence similarity with HIF-1α. They hetero-dimerize with HIF-1β, and are regulated in a similar O_2_-dependent way with HIF-1α. However, whereas HIF-1α is expressed in all nucleated cells in most of the metazoan species, HIF-2α and HIF-3α are only expressed in certain cell types in vertebrate species [19]. It was reported that HIF-1α is rapidly activated under acute, severe hypoxia (1–2% O_2_), or anoxia (<0.1% O_2_) during short periods (2–24 h) in some cell lines, whereas HIF-2α gradually accumulates under mild or physiological hypoxia (<5% O_2_), and sustains for a long time (48–72 h) [91,92]. HIF-1α and HIF-2α appears to play opposite but balancing roles during the hypoxic response under both physiological and pathophysiological conditions in some cells. The function of HIF-3α, which lacks the transactivation domain, is not clear in detail. HIF-3α has many splice variants [93]; some variants appear to activate gene expression [94,95], while others might be negative regulators of HIF-1α and HIF-2α [96,97,98].

Through transcriptional regulation of different genes by HIFs and other mechanisms, hypoxia regulates many different cell processes, (1) to change the hypoxic environment (e.g., inducing angiogenesis), (2) to reprogram the cellular metabolism to adapt to hypoxia (e.g., increase in glycolysis), (3) to let cells to move to a new place without hypoxia (e.g., promoting cell migration and invasion), and eventually resulting in cell death or survival [19,21]. As an oxidative-related and iron-dependent regulated cell death, ferroptosis is an important form of cell death involved in the hypoxia-induced cell response.

## 4. The Mechanism and Regulation of Ferroptosis Modulated by Hypoxia

### 4.1. Ferroptosis Inhibited by Hypoxia and HIFs

#### 4.1.1. Hypoxia and HIFs Inhibit Ferroptosis in Tumor Cells

Hypoxia is often found during tumor progression, which is always accompanied by decreased cell death. Many studies reported that HIF-1 can inhibit the many kinds of regulated cell death, such as apoptosis and autophagy [99,100]. In addition to apoptosis and autophagy, ferroptosis was also found to be inhibited in tumors (Figure 2). In clear cell renal cell carcinoma (ccRCC), both HIF-1α and HIF-2α are stabilized by VHL loss, leading to the decrease in ferroptosis sensitivity to erastin or BSO via fatty acid β-oxidation and mitochondrial ATP-synthesis [101]. A recent study on clear cell renal cell carcinoma showed that the iron sulfur cluster assembly 2 (ISCA2), a component of the late mitochondrial iron sulfur cluster assembly complex, was induced by hypoxia (1% O_2_) to promote HIF-1α/HIF-2α translation, leading to the inhibition of iron overload and erastin- or RSL3-induced ferroptosis [102].

Therefore, ISCA2 inhibition induces ferroptosis and reduces tumor growth in vivo. In malignant mesothelioma under 1% O_2_ hypoxia, HIF-1 induces *Carbonic anhydrase 9* (*CA9*) expression, which in turn reduces catalytic Fe^2+^ by downregulating TFRC and upregulating FTL and FTH1 to inhibit erastin-induced ferroptosis [103]. In human fibrosarcoma HT-1080 cells and non-small cell lung cancer Calu-1 cells, HIF-1α stabilized by 1% O_2_ hypoxia enhances fatty acid uptake and lipid storage by transcriptionally elevating *FABP3* and *FABP7* expression, and finally inhibits RSL3-induced ferroptosis [59]. In gastric cancer cells under the 1% O_2_ hypoxia condition, HIF-1α transactivates the long noncoding RNA *CBSLR*, which in turn forms a complex with YTH N6-methyladenosine RNA binding protein 2 (YTHDF2) and *CBS* mRNA to reduce *CBS* mRNA stability in an N6-methyladenosine (m^6^A) modification-dependent manner, thus reducing the ACSL4 methylation, degrading ACSL4 via the ubiquitination-proteasome pathway, and eventually protecting gastric cancer cells from ferroptosis in a hypoxic tumor microenvironment [104]. Another study on the peritoneal metastasis of gastric cancer showed that 1% O_2_ hypoxia activates HIF-1α to transcriptionally induce *lncRNA-PMAN* expression via binding to the HRE sequence in its promotor, which recruits the embryonic lethal abnormal vision like RNA binding protein 1 (ELAVL1) to cytoplasm to stabilize *SLC7A11* mRNA, thus leading to the inhibition of erastin- or RSL3-induced ferroptosis [105]. The *SLC7A11* mRNA is also stabilized by blocking its N6-methyladenosine (m^6^A) modification and YTHDF2-dependent degradation by suppressing methyltransferase like 14 (METTL14), which is a central component of the m6A methylated transferase complex, in a HIF-1α-dependent manner, to inhibit ferroptosis in hepatocellular carcinoma [106]. Furthermore, it was reported that hypoxia (1% O_2_) induces HIF-1α through activating PI3K to phosphorylate AKT, then upregulating *SLC7A11* expression in glioma cells to suppress sulfasalazine-induced ferroptosis [107]. The core circadian clock gene period 1 (PER1) can bind to and degrades HIF-1α; while HIF-1α can bind to the *PER1* promotor and reduce *PER1* transcription, which forms a negative feedback loop [108]. In oral squamous cell carcinoma, PER1 expression is often decreased, which in turn induces HIF-1α to inhibit ferroptosis via promoting *GPX4* and *SLC7A11* expression and inhibiting *TFRC* expression [108]. A recent study showed that HIF-1α stabilized by myriocin, an inhibitor of the de novo synthesis of sphingolipid, could promote the expression of *PDK1* (3-phosphoinositide-dependent protein kinase-1, an enzyme involved in glycolysis) and *BNIP3* (BCL2/adenovirus E1B 19 kDa protein-interacting protein 3, a member of the apoptotic Bcl-2 protein family) and alter the intracellular levels of glucose metabolites, to inhibit erastin- or glutamate-induced ferroptosis in HT-1080 human fibrosarcoma cells, but not in GES-1 human gastric epithelial cells and SK-Hep-1 human hepatoma cells [109]. Taken together, HIFs, especially HIF-1α, play a crucial role in suppressing ferroptosis by hypoxia.

In addition to the HIFs, hypoxia also inhibits ferroptosis of tumor cells through other mechanisms. In HT-1080 fibrosarcoma cells, 1% O_2_ hypoxia increases FTH1 expression in an NCOA4-independently manner to inhibit RSL3-induced ferroptosis [110]. In non-small cell lung cancers, 1% O_2_ hypoxia induces the intracellular expression of angiopoietin-like 4 (ANGPTL4) and its extracellular secretion by exosomes to neighboring normoxic cells, both of which increase the expression of GPX4, SLC7A11, FTL, and FTH1, to inhibit the RSL3- or irradiation-induced ferroptosis [111].

#### 4.1.2. Hypoxia and HIFs Inhibit Ferroptosis in Normal Cells

Hypoxia and HIFs suppresses ferroptosis not only in tumor cells but also in normal cells. In primary human macrophages, hypoxia (1% O_2_) reduces NCOA4 expression to inhibit ferritinophagy, resulting in the increases in FTMT levels to reduce RSL3-induced ferroptosis [110]. A following study showed that hypoxia (1% O_2_) stabilized HIF-2α to promote the transcription of *FTMT*, and activated thrombin to cleave FTMT from a 27 kDa precursor to a 22 kDa mature form, both of which resulted in increased FTMT to attenuate erastin- or RSL3-induced ferroptosis [112]. In the receptor activator of nuclear factor Kappa B ligand (RANKL)-induced differentiation of osteoclasts, hypoxia inhibits RANKL-induced ferritinophagy via inhibiting autophagosome formation by HIF-1α, thereby protecting osteoclasts from ferroptosis [113]. In rat embryonic cardiomyoblast H9C2 cells, 1% O_2_ hypoxia upregulates SENP1 to promote the deSUMOylation of HIF-1α and ACSL4, leading to HIF-1α stabilization and decreased ACSL4 protein levels, in turn protecting cells against erastin-induced ferroptosis [114]. In pulmonary artery smooth muscle cells and a pulmonary arterial hypertension rat model, hypoxia (3% O_2_ for cells and 10% O_2_ for mice, respectively) induces the deubiquitinase OTU domain-containing ubiquitin aldehyde-binding protein 1 (OTUB1), to stabilize SLC7A11, and eventually inhibiting erastin-induced ferroptosis [115]. It was reported recently that hypoxic-ischemic conditions induce HSPB1 in the hippocampus tissues to repress the hypoxia (anoxia)-induced ferroptosis of neuronal cells and hypoxic-ischemic brain damage via promoting expression of glucose-6-phosphate dehydrogenase, GPX4, and SLC7A11, as well as augmenting TFRC levels [116]. HIF-1α stabilized by myriocin was also found to inhibit erastin-induced ferroptosis via promoting *PDK1* and *BNIP3* expression as well as altering the intracellular glucose metabolism in HT22 mouse hippocampal neuronal cells and PC-12 rat neural cells [109]. Another recent study in cardiomyocytes showed that miR-210-3p is enriched in the hypoxia (1% O_2_)-conditioned cardiac microvascular endothelial cells-derived exosomes, which inhibits *TFRC* expression by directly interacting with *TFRC* mRNA, attenuating erastin-induced myocardial cell ferroptosis [117].

### 4.2. Ferroptosis Promoted by Hypoxia and HIFs

#### 4.2.1. Hypoxia and HIFs Promote Ferroptosis in Tumor Cells

Ferroptosis is not always inhibited by hypoxia and HIFs. In tumor cells, hypoxia and HIFs also induces ferroptosis in some conditions (Figure 3). Interestingly, HIF-2α, but not HIF1α, mainly mediates the induction of ferroptosis in tumor cells. In renal clear-cell carcinomas, HIF-2α activates *HILPDA*, which selectively enriches PUFA to promote lipid peroxidation and GPX4 inhibitor (RSL3, ML210, or ML162)-induced ferroptosis [61]. In glioblastoma (GBM), HIF-2α induced by the prolyl hydroxylase (PHD) inhibitor roxadustat, upregulated ferroptosis regulatory genes such as *ACSL4*, *PTGS2*, and *CHAC1*, to enhance lipid peroxidation and erastin-induced ferroptosis, leading to the suppression of GBM cell growth in vitro and in vivo [118]. Another report showed that HIF-2α upregulated genes involved in lipid and iron metabolism in both colorectal cancer (CRC) cells and colon tumors in mice, which result in the cells being susceptible to ferroptosis induced by dimethyl fumarate [119]. It was also reported that lysine (K)-specific demethylase 4A (KDM4A), which was highly expressed in cervical cancer tissue and could be induced under cobalt chloride mimicking hypoxia conditions, reduces the H3K9me3 level in the *HIF-1α* promoter region to elevate *HIF-1α* transcription, leading to the increased expression of *TFRC* and *DMT1* via activating the HRE sequence in their promoters, and finally resulting in the increase in erastin-induced ferroptosis of cervical cancer cells [120]. In addition to HIFs, hypoxia also induced E2F transcription factor 7(E2F7) to increase the transcription of splicing factor *quaking* (*QKI*), which promotes the biogenesis of *circBCAR3*, a circular RNA highly expressed in the esophageal cancer cells [121]. *CircBCAR3* binds to *miR-27a-3p* by the competitive endogenous RNA mechanism to upregulate *transportin-1* (*TNPO1*), a binding partner of CA9, and leading to the inhibition of CA9 to promote ferroptosis.

#### 4.2.2. Hypoxia and HIFs Induce Ferroptosis in Normal Cells

Ferroptosis can also be induced in normal cells under hypoxia conditions. It was reported that chronic intermittent hypoxia (CIH) induced ferroptosis in the hippocampus, lung, liver, and cardiomyocytes by downregulating NRF2 and GPX4, as well as upregulating ACSL4, leading to cognitive impairment and injury of brain, lung liver, and heart [122,123,124,125]. In the brain, hypoxic-ischemic induces ferroptosis with the increased TFRC expression, and decreased expression of SLC7A11, TRX-1, and GPX4 [126,127,128]. A recent study showed that hypoxia and ischemia induce chromobox7 (CBX7) in neural progenitor cells, which promote ferroptosis through suppressing the NRF2/HO-1 signaling pathway [128]. In microglial BV-2 cells, hypoxia down-regulates GPX4 and SLC7A11 to induce ferroptosis, which can be reversed by the wild bitter melon extract [129].

It was also found that the hypoxia (1% O_2_) increased miR-30b-5p to repress SLC7A11 and Pax3 (a transcription factor that promotes the expression of *SLC7A11* and the iron exporter *FPN1*) levels, to induce ferroptosis of trophoblasts, leading to preeclampsia [130]. Additionally, this hypoxia-induced ferroptosis of trophoblasts in the placenta can be inhibited by the phospholipase PLA2G6, which catalyzes the hydrolysis of oxidized PUFAs from membrane phospholipids to attenuate ferroptosis [62]. On the contrary, miR-2115-3p interacts with the mRNA of *glutamic-oxaloacetic transaminase1* (*GOT1*) to repress its expression, increasing GPX4 levels and decreasing ACSL4 and TFRC levels, and eventually leading to inhibiting the hypoxia-promoted ferroptosis in a preeclampsia model [131].

In retinal pigment epithelium cells, stabilized by the 3%O_2_ hypoxia condition and PHD inhibitor dimethyloxalylglycine (DMOG) treatment, HIF aggravates sodium iodate-induced ferroptosis by upregulating *superoxide dismutase* (*SOD*) to execute a peroxidative rather than antioxidative role, as well as increasing the iron importers *ZIP8* and *ZIP14* to enhance iron import [132]. Moreover, HIF-2α was found to be upregulated by myostatin in response to cigarette smoke exposure, leading to ferroptosis in myotubes and chronic obstructive pulmonary disease-related skeletal muscle dysfunction [133].

As an environment featuring hypobaric hypoxia, high altitude (HA) exposure increased ferroptosis sensitivity in adipose tissue with elevated levels of iron, ROS, MDA, and 4-HNE, as well as GSH depletion [134]. Acute high-altitude hypoxia exposure in mice leads to cerebral formaldehyde accumulation to induce neuronal ferroptosis [135]. A recent study suggested that hypobaric hypoxia at high-altitude induces both apoptosis and ferroptosis via the JNK signaling pathway by depleting keratin 18 (Krt18) and elevating JNK3 (MAPK10), as well as increasing *ASCL4* for ferroptosis [136]. An integrative comparison of the oviduct epithelial cells between yak at high altitude and bovine suggested that the mitophagy-animal pathway and HIF-1 signaling pathway may also be involved in high-altitude-induced ferroptosis [137].

#### 4.2.3. Hypoxia/Reoxygenation (H/R) Promotes Ferroptosis in Normal Cells

I/R in organs, which leads to the H/R of cells, often results in injury because of cell ferroptosis. Many studies reported ferroptosis induction in IRI, especially in the heart. A study on myocardial IRI reported that H/R represses NRF2 to reduce *FPN1*, resulting in the increase in Fe^2+^ and erastin-induced ferroptosis in H9C2 cardiomyocytes [138]. H/R also induces *PTGS* through HIF-1α to induce ferroptosis in H9C2 cardiomyocytes [139] and a rat model of coronary microembolization (CME)-induced myocardial injury [140]. H/R treatment decreased *SMAD7* expression and increased *Hamp1* expression to promote erastin-induced ferroptosis in H9C2 cardiomyocytes [141]. Atorvastatin blocked the HIF-1α/COX-2 axis and the SMAD7/hepcidin pathway to inhibit CME- or erastin-induced ferroptosis of cardiomyocytes [140,141]. It is reported that H/R treatment induces USP7 to activate p53, leading to the increased TFRC levels to promote ferroptosis of H9C2 cardiomyocytes, as well as in a rat model of myocardial IRI [142]. DNA (cytosine-5)-methyltransferase 1 (DNMT-1) was found to augment the H/R-induced ferroptosis of H9C2 cardiomyocytes by enhancing NCOA4-mediated ferritinophagy by elevating DNA methylation in the *NCOA4* promoter [143]. DMT1 expression is significantly induced by H/R treatment to promote the ferroptosis of myocardial cells isolated from mouse models for acute myocardial infarction (AMI) and cardiomyocyte hypoxia injury, which could be inhibited by miR-23a-3p carried by the exosome from human umbilical cord blood-derived mesenchymal stem cells [144]. The transcription factor forkhead box C1 (FOXC1) transcriptionally elevates the expression of ELAVL1, which binds to and stabilizes *Beclin-1* mRNA, to promote the ferroptosis of cultured myocardial cells exposed to H/R and mouse myocardial I/R model [145]. A recent study demonstrated that HO-1 is upregulated in response to hypoxia (0.5% O_2_) and H/R to degrade heme, thereby resulting in iron overload and ferroptosis in the endoplasmic reticulum (ER) of cardiomyocytes [146]. Moreover, ferroptosis is also induced by activating endoplasmic reticulum stress in H/R-treated H9C2 cardiomyocytes and cardiomyocytes in the myocardial IRI model [147].

In addition to the heart, ferroptosis was also found in the IRI of other organs. A study on lung IRI showed that H/R induces lung epithelial cell ferroptosis through enhancing ACSL4 expression [148]. In intestinal I/R-induced acute lung injury (ALI), HIF-1 activation by H/R increases the mRNA levels of *TF*, *PTGS2*, and *ACSL4* to promote erastin- or H/R-induced ferroptosis of mouse lung epithelial (MLE)-2 cells, which can be blocked by isoliquiritin apioside or by enhancing iASPP (inhibitor of apoptosis-stimulating protein of p53)/NRF2 [149,150]. A study on IRI in the intestine also demonstrated that special protein 1 (Sp1), a crucial transcription factor, was increased in the hypoxic condition (1% O_2_) or ischemia to promote *ACSL4* expression, thus enhancing H/R-induced ferroptosis of cells and I/R-induced ferroptosis in mouse intestine [151]. H/R induces the expression of transmembrane member 16A (TMEM16A), a component of the hepatocyte Ca^2+^-activated chloride channel, which interacted with GPX4 to induce its ubiquitination and degradation, thereby enhancing the erastin- or RSL3-induced ferroptosis of hepatocytes, contributing to the I/R-induced liver injury [152]. In IRI-induced acute kidney injury (AKI), H/R induces the ferroptosis of human proximal tubular epithelial cells (HK-2) through upregulating ACSL4 and COX-2, as well as down-regulating GPX4 and FTH1 in the kidney tissues [153]. Either decreasing ACSL4 by blocking serine/arginine splicing factor 1 (SRSF1) with lncRNA TUG1 carried by urine-derived stem cells (USCs)-derived exosomes (USC-Exo), or inhibiting the activation of the c-Jun NH2-terminal kinases (JNK) pathway by inositol requiring enzyme 1 (IRE1), a proximal ER stress sensor, can inhibit the H/R-induced ferroptosis of HK-2 cells [153,154]. Additionally, ubiquitin-specific protease 11 (USP11) is elevated in neuronal cells after H/R and in the spinal cord in mice with IRI, which promotes erastin-induced autophagy-dependent ferroptosis by stabilizing Beclin-1 [155].

In most studies on IRI, the cells were treated by H/R or in organs with I/R. It is difficult to determine whether the promotion of ferroptosis resulted from hypoxia/ischemia or reoxygenation/reperfusion. As the reoxygenation process may increase oxidative stress, it is often considered the reason for H/R-induced ferroptosis. However, studies that employed both hypoxia and H/R treatment found that hypoxia alone could promote ferroptosis, which could not be further elevated by reoxygenation in some cases [146,151]. Another study focused on the reoxygenation process showed that reoxygenation did not alter Nrf2 or HIF-1α activity [156], whereas these two pathways have been reported to be involved in the H/R treatment in several studies [138,139,140,141,149,150]. These studies suggested that hypoxia/ischemia at least partially contributes to promoting ferroptosis. Therefore, we also reviewed the ferroptosis promoted by H/R in this part, although we cannot attribute all the effects on ferroptosis to hypoxia.

### 4.3. Other Mechanisms Involved in Ferroptosis Regulated by Hypoxia

Besides the genes that directly mediate the regulation of ferroptosis by hypoxia and HIFs, some genes that are not induced by hypoxia, modulate the ferroptosis regulated by hypoxia through different mechanisms.

For example, a study on tumor recurrence found that hypoxia during treating cancer cells with tyrosine kinase inhibitors or cisplatin, which is confirmed by CA9 elevating as a hypoxia marker, induced SCD1 expression in cancer cells, as well as FABP4 in tumor endothelial cells (TECs) and adipocytes in the tumor microenvironment (TME), to inhibit ferroptosis and promote tumor recurrence [58]. Mechanistically, SCD1 catalyzes the fatty acid desaturation to produce monounsaturated fatty acids (MUFA), leading to ferroptosis inhibition in cancer cells, while FABP4 in the TME sustains lipid droplet (LD) formation and promotes cancer cell survival under hypoxia-induced ferroptosis. Additionally, diabetes generates NOX2 in an AMPK-dependent manner to enhance oxidative stress, which led to promoting H/R-induced ferroptosis in H9C2 cardiomyocytes [157]. MiR-124-3p, which is enriched in the HO-1-modified bone marrow mesenchymal stem cells-derived exosomes, reduces *STEAP3* by directly binding to its mRNA, to attenuate the H/R-induced ferroptosis of IAR20 (normal rat hepatocyte cell line) and LO2 (human fetal hepatocyte cell line) by changing iron homeostasis [158]. In acute spinal cord injury (ASCI), lncGm36569 carried by mesenchymal stem cells-derived exosomes (MSCs-exo) can act as a competitive RNA of miR-5627-5p to induce FSP1 expression, thereby attenuating hypoxia-induced ferroptosis neuronal cell and neuronal dysfunction [159].

There are also some drugs regulating hypoxia-induced ferroptosis through the mechanisms independent of hypoxia. In myocardial IRI, dexmedetomidine activates Nrf2 via AMPK/GSK-3β (AMP-activated protein kinase/Glycogen synthase kinase 3β) pathway, to inhibit H/R induced-ferroptosis of cardiomyocytes and to protect hearts from IRI [160]. It was also reported that icariin and irisin protect against myocardial and lung injury, respectively, by suppressing H/R-induced ferroptosis of cardiomyocytes and lung epithelial cells via activating Nrf2/HO-1 signaling [161,162]. Additionally, dimethyl fumarate, a therapeutic agent for relapsing-remitting multiple sclerosis, inhibits H/R-induced ferroptosis in alpha mouse liver cells and a mouse liver IRI model by activating NRF2 pathway [163]. Melatonin, an antioxidant that regulates the sleep-wake cycle, was reported to inhibit RSL3-, erastin-, or H/R-induced ferroptosis through modulating NRF2, AKT, and GPX4, in hypoxic-ischemic brain damage, as well as through upregulating NRF2 and downregulating SLC7A11 in mouse tubular epithelial cells and a mouse acute kidney injury (AKI) model [164,165]. Moreover, gastrodin, one of the main functional substances of functional food raw material *Gastrodia elata BI*, inhibited hypoxia- or erastin-induced ferroptosis in hippocampal neurons by activating NRF2 and GPX4, as well as inhibiting KEAP1 and COX-2 [166]. In hypoxic-ischemic brain damage, glycyrrhizin (GL), an HMGB1 inhibitor, can suppress the RSL3- or oxygen-glucose-deprivation-induced neuronal ferroptosis via the HMGB1/GPX4 pathway [167]. A recent study reported that Xingnaojing, a traditional Chinese medicine, inhibits ferroptosis in hypoxia-treated SH-SY5Y neuroblastoma cells and middle cerebral artery occlusion (MCAO)-induced cerebral ischemia rats, via upregulating HO-1, FPN, GPX4 and downregulating TFRC, DMT1, and COX-2 [168]. Additionally, lidocaine, a local anesthetic, was found to attenuate the H/R-induced ferroptosis of lung epithelial cells in lung IRI via the p38 MAPK pathway [169].

Although they are not dependent on hypoxia clearly, all these mechanisms are also involved in the regulation of ferroptosis by hypoxia.

## 5. Conclusions and Perspectives

The effect of hypoxia and HIFs on ferroptosis is highly context-dependent, especially for determining cell death or survival. According to the studies we reviewed above, hypoxia has a dual role in both inhibiting and promoting ferroptosis in a context-dependent manner. Hypoxia’s effect on ferroptosis seems to mainly depend on the cell type: hypoxia usually inhibits ferroptosis in cancer cells, while in normal cells, hypoxia often induces or promotes ferroptosis. However, there is some exception: hypoxia was reported to promote ferroptosis in certain cancer cells; and in several normal cells, hypoxia was found to inhibit ferroptosis. These suggested that the regulation of ferroptosis by hypoxia may depend on the expression of other genes involved in ferroptosis, which will be differently expressed in tumor and normal cells. Studying the cancer cells and normal cells simultaneously may be helpful to understand the different regulations. However, most studies only focus on the regulation of ferroptosis by hypoxia in either tumor or normal cells. Only a few studies simultaneously compare them. Fuhrmann et al. found that hypoxia protects both primary human macrophages and HT-1080 fibrosarcoma cells from RSL3-induced ferroptosis, through increasing FTH1 and FTMT in macrophages, and through increasing only FTH1 but not FTMT in HT-1080 cells that basally express FTH and low levels of FTMT [112]. Another study by Liu and colleagues reported that the sphingolipid synthesis inhibitor myriocin stabilized HIF1α to decrease erastin-induced ferroptosis in two normal cell lines (mouse hippocampal neuronal cell HT22 and rat neural cell PC-12) and a tumor cell line (human fibrosarcoma cell HT-1080), but not in another normal cell line (human gastric epithelial cell GES-1) and tumor cell line (human hepatoma cell SK-Hep-1) [109]. However, they did not further investigate the different regulations of ferroptosis in these cell lines. As these two studies only investigated two tumor cell lines from different tumor types and compared them with the normal cells from different tissues and organs, it is hard to find out whether the difference of ferroptosis regulation by hypoxia is dependent on some genes differently expressed in tumor and normal cells.

Interestingly, hypoxia also has a dual effect on another form of regulated cell death, apoptosis, which is apparently dependent on the oxygen concentration: oxygen levels in the range of 0–0.5% (anoxia) in cells induce apoptosis, whereas oxygen levels in the range of 1–3% (hypoxia) in cells do not lead to apoptosis but protect cells for survival [170,171]. From the studies we reviewed, hypoxia-regulated ferroptosis does not appear to be related to the oxygen concentration. A potential reason for this difference is the oxidative stress induced by hypoxia. Oxidative stress always induces ferroptosis, whereas apoptosis is only induced in severe oxidative stress, and mild oxidative stress leads to cell cycle arrest and survival [172,173]. Thus, different oxygen concentrations lead to different levels of oxidative stress, resulting in the controversial effects on apoptosis, as well as similar effects on ferroptosis. It is also worth to notice that most above-mentioned studies on ferroptosis used oxygen concentration at 1–3% for the hypoxic conditions, which could induce not apoptosis but only ferroptosis. Hypoxia-regulated ferroptosis in these studies may be independent of apoptosis. Only a few studies on IRI used <1% O_2_ or anoxic conditions to mimic I/R, leading to ferroptosis in normal cells, which may also induce apoptosis.

Mechanistically, hypoxia inhibits ferroptosis mainly by activating HIF-1α, whereas it promotes ferroptosis mainly by activating HIF-2α, in both normal and cancer cells. Studies also reported that HIF-1α mediates the hypoxia-induced ferroptosis in normal cells in certain conditions, and HIF-2α mediates the inhibition effect of hypoxia on ferroptosis in some cancer cells. The roles of HIF-1α and HIF-2α in ferroptosis appear to be different from previous studies, which suggests that HIF-1α plays a key role in the initial response to acute and intense hypoxia, whereas HIF-2α drives the hypoxic response during chronic and mild hypoxic exposure [91,92,174]. Although HIF-2α still mediates some effects of chronic hypoxia on ferroptosis, it is also involved in the acute effects of hypoxia. Moreover, non-canonical mechanisms modulating HIFs were also found in the regulation of ferroptosis by hypoxia. It was also reported that HIF-1α and HIF-2α have different targets: HIF-1α regulates most of the glycolytic-related genes, whereas HIF-2α selectively regulates many iron-regulatory genes [175,176]. Consistently, the abovementioned studies showed that HIF-2α inhibits ferroptosis through its direct targets involved in iron metabolism, such as *FTMT*, in both tumor and normal cells [102,112]. However, it is also reported that HIF-1α regulates iron metabolism directly through transcriptionally regulation of its targets, such as *TFRC* and *DMT1* [120], as well as indirectly through some other targets, such as *CA9* [103]. Fe^2+^ is a cofactor for PHDs and FIH-1, which regulates HIFs under normoxia. Therefore, the regulation of iron metabolism by HIF-1α and HIF-2α seems to be very complex, and cannot be summarized here. In addition to HIFs, NRF2, the master regulator of oxidative stress signaling, also plays an important role in hypoxia-induced ferroptosis. Many studies reported that NRF mediates the regulation of ferroptosis induced by hypoxia or H/R [124,125,128,138], while another group of studies reported that targeting NRF2 could inhibit hypoxia-or H/R-induced ferroptosis [149,160,161,162,163,164,165,166].

Hypoxia-regulated ferroptosis is involved in many diseases, such as cancer and IRI of the organs. Thus, understanding the mechanism for hypoxia to regulate ferroptosis may contribute to developing the therapy for these diseases. For instance, as hypoxia is a biomarker of solid tumors, targeting the hypoxic area is often used as a strategy to deliver drugs to tumor cells, including drugs to induce ferroptosis [177,178,179]. Since hypoxia inhibits ferroptosis in many tumor cells, the delivery of ferroptosis inducers to the hypoxic area may limit their effect on cancer cells. From understanding the mechanism involved in the inhibition of ferroptosis caused by hypoxia, avoiding the ferroptosis inhibition caused by hypoxia could be a new strategy for treating tumors with ferroptosis inducers [180,181]. Considering that ferroptosis is the major reason for IRI, many drugs have been developed to inhibit hypoxia-induced ferroptosis based on its mechanism [129,140,141,150,160,161,162,163,164,165,166,167,168,169]. Intriguingly, hypoxia is a symptom of the Coronavirus Disease-2019 (COVID-19); HIF-1α plays an important role in the early phase of SAR-CoV-2 infection and is also associated with secondary organ damage [182,183]. On the other hand, some clinical features in COVID-19 pathobiology, such as erythropoiesis suppression and anemia, were found to result from the dysregulation of iron homeostasis and abnormal ferroptosis [183]. However, few studies reported a direct connection between hypoxia and ferroptosis in COVID-19. Further investigation of the effect and mechanism regulating ferroptosis caused by SAR-CoV-2 infection-induced hypoxia could provide more strategies for treating COVID-19 or the post-acute sequelae of SARS-CoV-2 infection through ferroptosis inhibition.

Taken together, hypoxia regulates ferroptosis through a complex network in a context-dependent manner, leading to the inhibition or promotion of ferroptosis in different cell types and conditions, which is mediated by HIFs, NRF2 signaling, and other mechanisms. The regulation of ferroptosis by hypoxia is involved in many diseases, such as cancer and the IRI of the organs. Therefore, elucidating its mechanism could be helpful to invent novel therapeutic strategies for these diseases.

## Figures and Tables

**Figure 1 cells-12-01050-f001:**
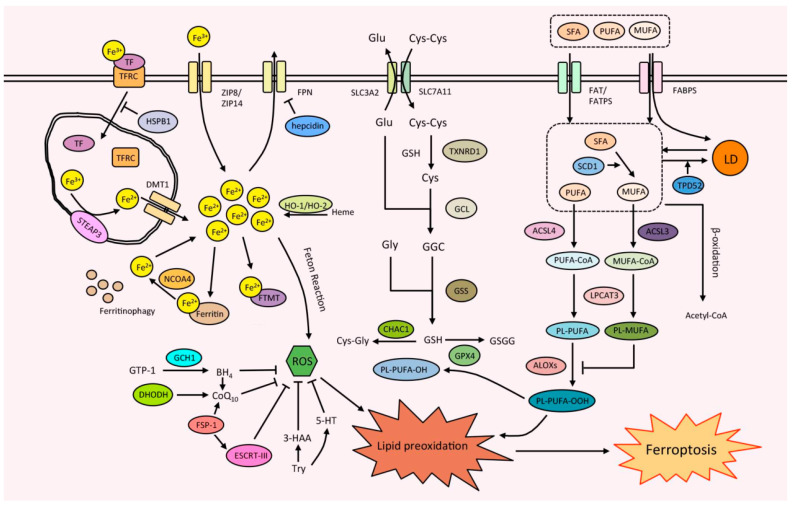
Ferroptosis is caused by the abnormal increased iron accumulation, lipid peroxidation, and dysregulation of antioxidant defense systems. There are complex regulations among iron metabolism, lipid metabolism, and the antioxidant defense system, and they cooperate with each other leading to lipid peroxidation and the occurrence of ferroptosis. Cys: cysteine; Cys-Cys: cystine; Cys-Gly: cysteinylglycine; Glu: glutamate; GGC: Gamma-Glu-Cys; GSSG: glutathione oxidized; LD: lipid droplet; PL: phospholipid; SFA: saturated fatty acid; and Try: tryptophan.

**Figure 2 cells-12-01050-f002:**
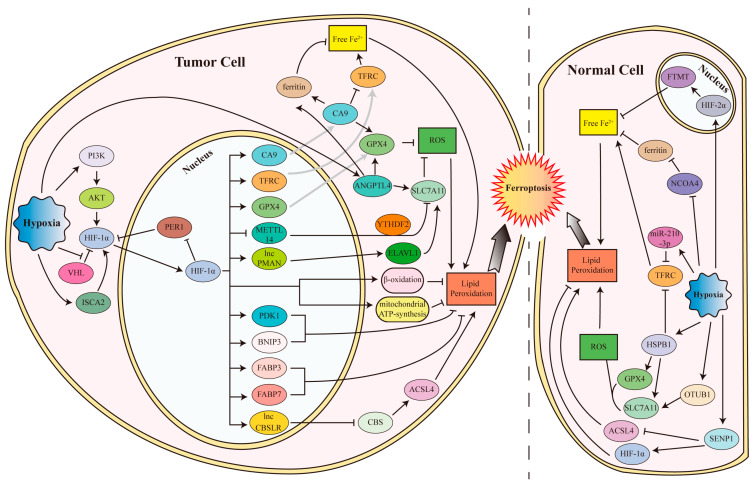
The mechanism of ferroptosis inhibited by hypoxia and HIFs in cells. In tumor cells, hypoxia activates HIF-1α in both canonical and non-canonical ways, to transcriptionally upregulate the expression of CA9, TFRC, GPX4, METTL14, PDK1, BNIP3, FABP3, FABP7, lncPMAN, and lncCBSLR, as well as increase protein levels of ANGPTL4, ferritin, SLC7A11, and GPX4, and decrease protein levels of TRFC and ACSL4, leading to the reduction in free Fe^2+^, ROS, and lipid peroxidation, and eventually inhibiting ferroptosis. In normal cells, hypoxia induces HIF-2α, HIF-1α, SENP1, OTUB1, HSPB1, and miR-210-3p, and represses NCOA4, to regulate FTMT, ferritin, TRFC, ACSL4, GPX4, SLC7A11, and ultimately inhibits ferroptosis. The gray arrows indicate the protein translation process from their corresponding mRNAs.

**Figure 3 cells-12-01050-f003:**
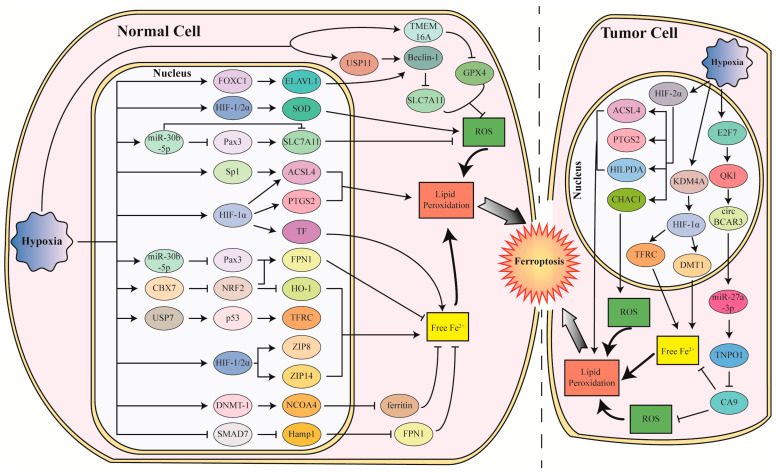
The mechanism of ferroptosis promoted by hypoxia, HIFs, or H/R in cells. In normal cells, hypoxia activates HIF-1α, HIF-2α, FOXC1, sp1, USP7, p53, CBX7, DNMT-1, USP11, TMEM16A, Beclin-1, and miR-30b-5p, and represses NRF2, SMAD7, and Pax3, to regulate SLC7A11, PTGS2, ACSL4, TF, FPN1, HO-1, TFRC, ZIP8, ZIP14, NCOA4, Hamp1, ELAVL1, SOD, GPX4, and ferritin, resulting in the induction of ferroptosis. In tumor cells, hypoxia regulates HILPDA, ACSL4, PTGS2, and CHAC1 by HIF-2α, TFRC, and DMT1 by HIF-1α and protein CA9 by E2F7/QKI/circBCAR/miR-27a-3p/TNPO1 to promote ferroptosis.

## Data Availability

Not applicable.

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
