# Peer review of "Ferroptosis Regulated by Hypoxia in Cells"

_cells, 2023, doi:10.3390/cells12071050_

Round 1

Reviewer 1 Report

In this review, the authors conclude that hypoxia plays a dual role in inhibiting and promoting ferroptosis in a context-dependent manner. The authors claim that the effect of hypoxia on ferroptosis is primarily cell type dependent. That is, hypoxia usually inhibits ferroptosis in cancer cells, whereas hypoxia often induces or promotes ferroptosis in normal cells. There are exceptions, however, where hypoxia has been reported to promote ferroptosis in certain types of cancer cells and to inhibit ferroptosis in some normal cells. These findings suggest that the regulation of ferroptosis by hypoxia may depend on the expression of other genes involved in ferroptosis, the expression of which may differ between tumor and normal cells. Specific comments are as follows.

Major points

1.     While I generally agree with the authors, they only use the relative number of papers to determine whether ferroptosis is inhibited by hypoxia in cancer cells and promoted in normal cells. The papers that test in cancer cells and the papers that test in normal cells are different papers and by different investigators. The authors should give weight to and properly cite papers that compare these phenomena (i.e., cancer cells and normal cells simultaneously) within a single paper.

2.     In the paper, the authors simply use the term ferroptosis, but it is important to note under what conditions ferroptosis was induced, as there are a variety of methods and drugs that induce ferroptosis.

3.     Since ischemia-reperfusion-induced ferroptosis is due to oxidative stress associated with reoxygenation, it is problematic that the authors seem to state that hypoxia promotes ferroptosis. Furthermore, these experiments typically examine the effects of hypoxia on normal organs, so it is not fair that the authors describe hypoxia as an example of hypoxia promoting ferroptosis in normal cells.

Minor points.

English should be carefully revised by a professional English editing service. 

Author Response

Dear Reviewer,

Thank you for your time and constructive comments on our manuscript. We have read your comments and addressed the critiques for this revision. We hope that you find our responses acceptable and approve this revised manuscript for publication. Following is our response to the comments.

Major points:

  1. While I generally agree with the authors, they only use the relative number of papers to determine whether ferroptosis is inhibited by hypoxia in cancer cells and promoted in normal cells. The papers that test in cancer cells and the papers that test in normal cells are different papers and by different investigators. The authors should give weight to and properly cite papers that compare these phenomena (i.e., cancer cells and normal cells simultaneously) within a single paper.

Thanks for the suggestion. We discussed and emphasized two studies that compared cancer cells and normal cells simultaneously in a single paper in the conclusion part of the revised manuscript.

  1. In the paper, the authors simply use the term ferroptosis, but it is important to note under what conditions ferroptosis was induced, as there are a variety of methods and drugs that induce ferroptosis.

Thanks for the suggestion. We mentioned the methods or drugs used to induce ferroptosis for reviewed studies in the revised manuscript.

  1. Since ischemia-reperfusion-induced ferroptosis is due to oxidative stress associated with reoxygenation, it is problematic that the authors seem to state that hypoxia promotes ferroptosis. Furthermore, these experiments typically examine the effects of hypoxia on normal organs, so it is not fair that the authors describe hypoxia as an example of hypoxia promoting ferroptosis in normal cells.

Thanks for pointing this out. We have isolated the studies on IRI into a separate subpart and described them in detail to clearly show the experiments of H/R-treated cells or I/R-treated organs in the revised manuscript. We also clearly state the experiments were using H/R or hypoxia alone. At the end of the subpart for IRI studies, we noted that the ferroptosis induced by H/R or I/R is not considered to be induced by hypoxia or ischemia, oxidative stress elevated by reoxygenation or reperfusion may play a more important role. Moreover, we mentioned two studies that employed both hypoxia and H/R treatment, which showed that hypoxia alone could promote ferroptosis independent of reoxygenation. We also pointed out that a study focused on the reoxygenation process showed that reoxygenation did not alter Nrf2 or HIF-1α pathways, which have been shown to mediate the effect of H/R or I/R in many studies. Therefore, hypoxia seems to contribute at least partially to H/R- or I/R-induced ferroptosis. And it is the reason we reviewed and summarized studies of IRI on ferroptosis in this subpart.

Minor points:

English should be carefully revised by a professional English editing service.

Thanks for the suggestion. We have our manuscript checked by the Language Editing Services recommended by the journal.

Again, we want to thank you for the constructive suggestions and great efforts to improve our manuscript. We hope that with these changes and response, our manuscript is acceptable for publication. 

Thank you very much!

Sincerely yours,   

Cen Zhang, Ph.D.

Professor,

College of Biological Science and Engineering,

Fuzhou University,

No. 2 Xueyuan Road

Fuzhou, Fujian 350108, China

Reviewer 2 Report

In their review article “Ferroptosis Regulated by Hypoxia in Cells” the authors summarize mechanisms of ferroptosis and hypoxia. The field of ferroptosis is fast growing and review articles are published frequently which includes a huge redundancy. This review article focuses on mechanisms and effects of hypoxia on ferroptosis which appears to be a unique feature. The article is well written, structured, and illustrated. In my opinion publishing this article in Cells would definitely support scientists in the fields of hypoxia and ferroptosis by giving a very comprehensive overview over the topics. I only have some minor comments which possibly could help the authors:

Line 51: surprising instead of suppressing?

Line 61: You may add scavengers for lipid peroxidation such as Liproxstatin-1

Line 418: In my opinion, cobalt chloride and also DMOG mimic hypoxia not completely but to a certain part (HIF-stabilization). Possibly the authors can rephrase a little bit.

Line 406: Would it make sense to make separate chapters for hypoxia and H/R and I/R?

Line 503-506: Please rephrase.

Line 565: The role of hypoxia for ferroptosis is highly context-dependent.

Author Response

Dear Reviewer,

Thank you very much for your careful review, and positive, insightful and constructive comments on our manuscript.  We have revised our manuscript accordingly and appropriate changes have been incorporated into our manuscript. We hope that you find our responses acceptable and approve this revised manuscript for publication. Following is our response to the comments.

Line 51: surprising instead of suppressing?

Thanks for pointing this out. You are right. We have corrected it in the revised manuscript.

Line 61: You may add scavengers for lipid peroxidation such as Liproxstatin-1

Thanks for the suggestion. We have added scavengers for lipid peroxidation (e.g., ferrostatin-1 or liproxstatin-1) into the revised manuscript.

Line 418: In my opinion, cobalt chloride and also DMOG mimic hypoxia not completely but to a certain part (HIF-stabilization). Possibly the authors can rephrase a little bit.

Thanks for pointing this out. We changed the subtitle to “4.1. Ferroptosis inhibited by hypoxia or HIFs” and “4.2. Ferroptosis induced by hypoxia or HIFs”

Line 406: Would it make sense to make separate chapters for hypoxia and H/R and I/R?

Thanks for the suggestion. We separate the H/R- and I/R-induced ferroptosis with the ferroptosis induced by hypoxia or HIFs in normal cells in a subpart of 4.2 in the revised manuscript.

Line 503-506: Please rephrase.

Thanks for the suggestion. We make some small changes in this sentence in the revised manuscript to emphasize that the aggravation of ferroptosis results from HIF stabilization by both hypoxia and DMOG treatment in this study.

Line 565: The role of hypoxia for ferroptosis is highly context-dependent.

Thanks for the suggestion. We changed the sentence to “The effect of hypoxia and HIFs on ferroptosis is highly context-dependent.”

Again, we want to thank you for the very nice comments, very constructive suggestions and great efforts to improve our manuscript. We hope that with these changes and response, our manuscript is acceptable for publication. 

Thank you very much!

Sincerely yours,   

Cen Zhang, Ph.D.

Professor,

College of Biological Science and Engineering,

Fuzhou University,

No. 2 Xueyuan Road

Fuzhou, Fujian 350108, China

Round 2

Reviewer 1 Report

The authors have addressed almost all of my concerns, and I have no further points.